# Multi-Temporal Passive and Active Remote Sensing for Agricultural Mapping and Acreage Estimation in Context of Small Farm Holds in Ethiopia

Tesfamariam Engida Mengesha [1,2], Lulseged Tamene Desta [3], Paolo Gamba [4,*] and Getachew Tesfaye Ayehu [3]

1   Department of Remote Sensing and Application Research and Development, Ethiopian Space Science and Geospatial Institute (SSGI), Entoto Observatory and Research Center (EORC), Addis Ababa P.O. Box 33679, Ethiopia; fishengida@gmail.com
2   Remote Sensing Department, Entoto Observatory and Research Center (EORC), Addis Ababa University, Addis Ababa P.O. Box 1176, Ethiopia
3   The Alliance of Bioversity International and CIAT, Addis Ababa P.O. Box 5689, Ethiopia
4   Telecommunications and Remote Sensing Laboratory, University of Pavia, 27100 Pavia, Italy
*   Correspondence: paolo.gamba@unipv.it

**Abstract:** In most developing countries, smallholder farms are the ultimate source of income and produce a significant portion of overall crop production for the major crops. Accurate crop distribution mapping and acreage estimation play a major role in optimizing crop production and resource allocation. In this study, we aim to develop a spatio–temporal, multi-spectral, and multi-polarimetric LULC mapping approach to assess crop distribution mapping and acreage estimation for the Oromia Region in Ethiopia. The study was conducted by integrating data from the optical and radar sensors of sentinel products. Supervised machine learning algorithms such as Support Vector Machine, Random Forest, Classification and Regression Trees, and Gradient Boost were used to classify the study area into five first-class common land use types (built-up, agriculture, vegetation, bare land, and water). Training and validation data were collected from ground and high-resolution images and split in a 70:30 ratio. The accuracy of the classification was evaluated using different metrics such as overall accuracy, kappa coefficient, figure of metric, and F-score. The results indicate that the SVM classifier demonstrates higher accuracy compared to other algorithms, with an overall accuracy for Sentinel-2-only data and the integration of optical with microwave data of 90% and 94% and a kappa value of 0.85 and 0.91, respectively. Accordingly, the integration of Sentinel-1 and Sentinel-2 data resulted in higher overall accuracy compared to the use of Sentinel-2 data alone. The findings demonstrate the remarkable potential of multi-source remotely sensed data in agricultural acreage estimation in small farm holdings. These preliminary findings highlight the potential of using multi-source active and passive remote sensing data for agricultural area mapping and acreage estimation.

**Keywords:** agriculture; optical and microwave remote sensing; machine learning

## 1. Introduction

The occurrence of climate variability, coupled with resource scarcity, social and political instability, pest outbreaks, and various other factors, has resulted in episodes of food insecurity in sub-Saharan Africa, placing the lives and livelihoods of the most disadvantaged communities at risk [1–4]. The agricultural system and landscape in sub-Saharan Africa are predominantly composed of smallholder farms that rely on rainfall to cultivate their crops [5–7]. Similarly, the agriculture system in Ethiopia is very complex, with 96% of the total cultivated area held by smallholders and producing a significant portion of the overall production for the major crops [8,9]. According to the Central Statistical Agency

(CSA), agricultural farms in Ethiopia are divided into two main categories: smallholder farms, which are farms with an area of less than 25.2 hectares, and large commercial farms, which are farms with an area of more than 25.2 hectares. The majority of farming systems in Ethiopia are smallholder farms, with the majority focused on subsistence agriculture, producing primarily for their consumption. In general, only 40% of smallholders cultivate more than 0.90 hectares, and these small-sized farms make up the majority of the total cultivated area in the country [9–11].

The agricultural mapping and estimation of crop production area of smallholder farms provides quantitative information for forecasting food security in communities [12]. Crop maps are important inputs for crop inventory production and yield estimation, and they can help farmers implement effective farm management practices and improve their livelihood [13]. However, there are no cropland extent maps at national and local scales in Ethiopia, particularly in the study area, that are regularly updated. The availability of digital crop extent map services may fill a gap in the country's current crop monitoring services by providing accurate, high-resolution, and regularly updated cropland area maps, as well as associated datasets [14].

The assessment of agricultural food production in Ethiopia, which relies on measuring crop area and crop yield, is particularly challenging due to the inadequacy and lack of accuracy in agricultural statistics, which is primarily attributable to inadequate organization and analysis of the data. Furthermore, agricultural statistics in Ethiopia are given at a coarse level, based on administrative units, affecting the accuracy and quality of the data [15,16]. Reliable information on where crops are grown, and their distribution patterns are essential for various purposes, including studying regional agriculture production, making informed political decisions, and enabling effective crop management. Accurately mapping agricultural crop distribution and estimating acreage play a significant role in optimizing crop production and resource allocation [17–20] and implementing and evaluating crop management strategies [21,22].

Remote sensing data are widely used for various applications in the agricultural domain, including soil property detection [23–25], crop type classification, and crop yield forecasting [26–30]. The information obtained from satellite images is dependent on the measurement of the electromagnetic energy reflected by different target features on the Earth's surface [31]. However, it is crucial to consider atmospheric effects, as they can significantly influence the reflectance values received by the satellite sensor [31,32]. Several studies have highlighted the challenge of differentiating between land surface target features with varying spectral signatures [33–35]. It has been observed that even similar land surface target features can exhibit different spectral signatures, making accurate classification challenging. Furthermore, the issue of similar reflection characteristics among different land cover classes in a study area adds complexity to the classification process, especially when working solely with optical images. In the past, satellite data used to analyze and understand agricultural lands were too generalized and limited in their ability to capture the diverse characteristics of these landscapes [36–38]. However, with the development of medium-resolution European Space Agency Sentinel constellation's products, there has been a significant improvement in the quality of spectral resolution as well as a substantial increase in spatial resolution. This enhancement in satellite capabilities has made it feasible to monitor smallholder farms in a more detailed and comprehensive manner [39–42].

While a single sensor's data may not be sufficient to optimize target class separation, incorporating radar data into classification models improves mapping accuracies due to increased cloud cover, independent data availability, and the physical and structural properties of the microwave signal—information that complements spectra from multispectral sensors [43]. The integration of radar imagery alongside optical images has proven to be beneficial [44–46]. Specifically, the integration of data from Sentinel-2 and Sentinel-1 allows for a more comprehensive understanding of complex landscapes in agricultural areas [44,47,48]. By incorporating radar data, the problems associated with different spectral signatures for similar land surface target features can be significantly

reduced. However, the integration of radar and optical imagery alone does not eliminate all issues. For example, the classification of water features can be problematic, as isolated water bodies might be mistakenly classified as bare land. Similarly, vacant lands can sometimes resemble agricultural or urban land cover classes, leading to misclassification. To overcome these specific challenges, researchers have explored the use of additional data sources, such as higher-resolution satellite imagery (e.g., a resolution of 5 m or finer). By exploiting these ancillary data sources, accurate training data can be obtained without the need for a field campaign, thus enhancing the separability between different land cover classes and improving classification accuracy.

Various studies have extensively discussed the challenges in remote sensing classification and have proposed various solutions: multi-source data fusion that includes the integration of radar and optical data, as well as the use of higher-resolution datasets and auxiliary sources for training data [43,49–52]. These approaches can solve problems related to varying spectral signatures and misclassification of land use land cover classes and improve the accuracy of the classification by employing advanced machine learning.

For agricultural mapping, numerous classifiers have been developed, with some of the most commonly employed being Support Vector Machines (SVM), Random Forest (RF), Gradient Tree Boosting (GTB), and Maximum Likelihood Classifiers [53–60]. Several studies have been conducted and tested to determine the most reasonable and accurate method among the machine learning classifiers used for LULC mapping [61–63]. Although the accuracy levels of each machine learning technique vary, it has been found that SVM and RF often provide more superior accuracy for classification than other classic classifier algorithms [64–67]. The major challenge in the application of these techniques for agricultural land mapping and acreage estimation is the lack of high spatial, temporal, and spectral information data. This is particularly important when considering smallholder farms [68].

Previous research has developed LULC maps with high spatial resolution at regional and local scales using commercial satellite data [69–71]. However, access to high-resolution, high-quality, and cloud-free satellite imagery is a challenge in many regions, particularly in the rainy season, which is the main agricultural season for Ethiopians. As a result, local planning agencies and governments lack adequate spatial information on smallholder farmers, which ultimately affects the monitoring of agricultural production and evaluation of the SDGs [12]. This is the reason for the selection of the study area in the Oromia regional state, which is known for its significant agricultural activity. The accurate delineation of agricultural land parcels and estimation of the areas provided by our approach will serve as important inputs for policymakers, agricultural planners, and land managers. The availability of such information supports targeted actions such as optimal resource allocation, land use planning, and crop yield prediction, thereby enhancing agricultural productivity and sustainability. The use of freely available multi-source imagery for agricultural mapping on small-scale farmlands is valuable for many developing countries with limited budgets for high-resolution data.

Therefore, by using the data from Sentinel-2 and Sentinel-1, this research aims to classify the study area into five common level-1 land use land covers and evaluate the potential of freely available sentinel products for mapping smallholder agricultural crop distribution and estimating acreage. In addition to investigating the suitability of sentinel products, the LULC classification techniques were tested to select the most accurate one for each investigated landscape [50,72,73]. The machine learning algorithms used in these experiments were RF, SVM, Classification and Regression Trees (CART), and GTB—chosen for their ability to discriminate between different classes, handle noisy data, and be applied with limited samples.

## 2. Materials and Methods

### 2.1. Study Area

The study was conducted in the Oromia region, Ethiopia, which is located geographically 7°32′45.736″ N and 40°38′4.866″ E. Oromia is one of Ethiopia's 12 regional states,

with the largest population and land area [74] (Figure 1). The region shares borders with several other regional states, including Amhara, Afar, Somali, Benishangul-Gumuz, Sidama, Southwest Region, and Central Region [75].

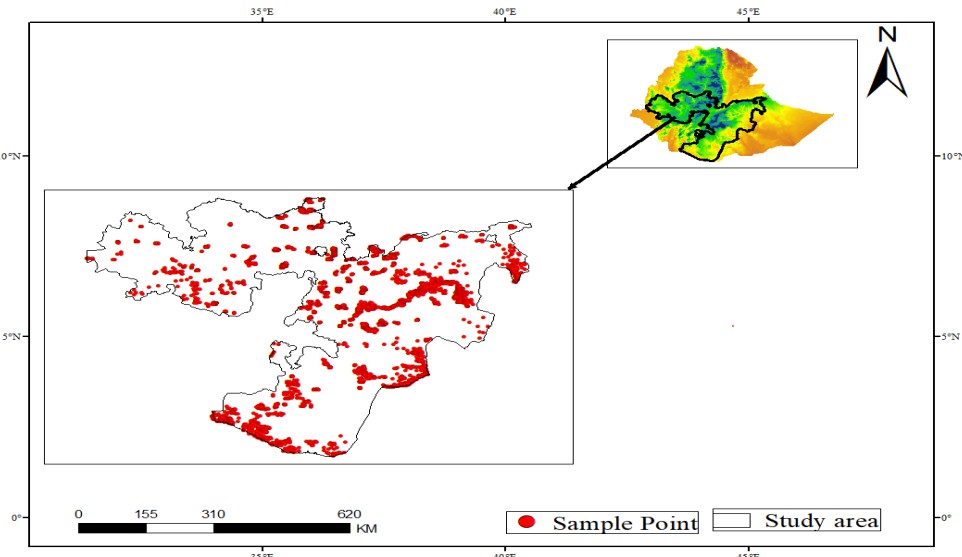

**Figure 1.** The boundary of the study area, red point is the sample point.

The Oromia region is characterized by a diverse and complex topography, which includes a variety of physiographic features that contribute to its unique landscape. These geographical features encompass mountains, rolling plateaus, river valleys, and plains area [76,77]. The study area climate is characterized by dry, tropical, and temperate climate zones with significantly varied amounts of annual precipitation between 410 and 2000 mm and temperatures between 18 and 39 °C [78].

It is the biggest crop-producing region, followed by Amhara and the South Nations and Nationality. The region's main crop is cereals, which cover 84% of the crop area [79]. In productivity area coverage and abundance, teff, maize, wheat, and sorghum are the most grown cereal grains. Also, this area grows a lot of vegetables, red pepper, Ethiopian cabbage, green pepper, and root crops like potato, sweet potato, and onion.

## 2.2. Methodology

The objective of performing Land Use and Land Cover (LULC) classification is to identify and extract particular land cover features from remotely sensed images. Although all surface features are not pertinent to our analysis, we adopted five distinct classes such as agricultural/crop, built-up/settlement, vegetation cover, bare-land, and water bodies, to avoid the binary classification of all surface features [80,81]. The selection of these classes is based on their relevance in understanding the overall agricultural landscape and distribution within the region. The effectiveness of the classifiers was then assessed quantitatively at the comprehensive class level, encompassing the entire study area depicted in Figure 1. Figure 2 illustrates a flow chart diagram describing the methodology.

In this study, two sets of images are created, one containing optical imagery and one created from the selected optical and SAR bands. Then, the bands and indices from Sentinel-2 and Sentinel-1 were combined into a single image. This can be achieved by stacking the relevant bands and indices along the band dimension.

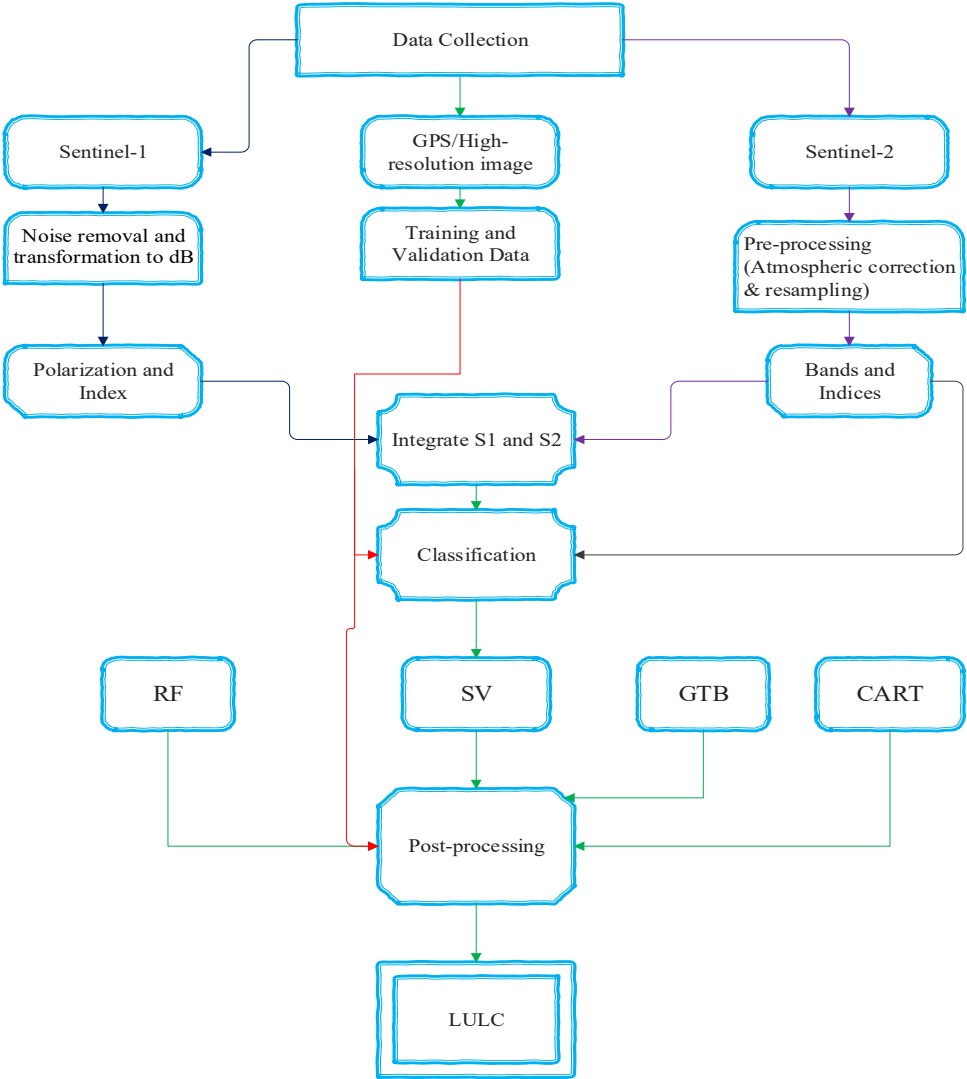

**Figure 2.** Methodology diagram.

To classify agricultural areas in a small and fragmented farmland landscape within the study area, the integration of Sentinel-1 and Sentinel-2 data was performed on the Google Earth Engine (GEE) cloud computing platform [82]. To determine the input variables for the classifications, the most informative multispectral bands related to the targeted features are filtered. Additionally, the spectral indices calculated from the two images of optical and microwave bands are created. Initially, the classification of the study area was carried out based on S-2 images. After that, S-1 data were retrieved over the same study period. S-1 data are spatially and temporally filtered. Following that, a stack of S-1 images is formed, resulting in a multiband multidimensional composite image rich in relevant information for later classification operations. This composite image consists of 4 bands that include both vertical and horizontal polarization components, their respective ratios, and a modified normalized ratio. Sentinel-1 vertical and horizontal polarization datasets have been integrated with the Sentinel-2 bands and indices. Furthermore, the collected images were reduced to a single image by computing their median values, resulting in a composite image for classification. Then, the second classification was performed based on the two integrated datasets at the GEE platform [83,84].

Then, the accuracy of each machine learning was assessed using different accuracy assessment metrics. Following the classification and accuracy assessments, the study area's agricultural land cover was quantified using an area estimation methodology. This involved determining the spatial extent of agricultural land within the study area by employing

pixel-based analysis and an error-adjusted area estimation by using the map class as the stratum of the sample.

### 2.2.1. Data Acquisition and Pre-Processing

Satellite Dataset

In this study, Sentinel-2 and Sentinel-1 satellite products were utilized for agricultural area mapping and acreage estimation. Sentinel-2 provides high-resolution multispectral optical imagery [85]. It consists of twin satellites that provide global coverage with a five-day revisit frequency at the equator, with 13 bands ranging from visible to short-wave infrared (SWIR) and varying spatial resolutions of 10–60 m [86]. Data from Sentinel-2 are processed at several levels. Level-1C (L1C) reflectance images are provided at the Top Of the Atmosphere (TOA), and Level-2A (L2A) reflectance images are provided at the Bottom Of the Atmosphere (BOA), derived from the L1C parameters [87].

The Sentinel-1 mission consists of two polar-orbiting satellites, Sentinel-1A, and Sentinel-1B, with C-band synthetic aperture radar instruments that operate day and night and can acquire imagery under any weather or illumination conditions [88–90]. Each Sentinel-1 satellite has a 12-day repeat cycle and provides C-band images in both solitary and dual polarization. Sentinel-1 data are openly accessible for enabling a variety of Earth monitoring applications because they are all-weather capable and have a high spatial resolution (up to 10 m) [81,91]. In this study, we employed the Sentinel-2 BOA data and the Sentinel-1 ground range detected (GRD) interferometric wide swath (IW) product.

Reference and Ground Truth Dataset

To establish reliable reference and ground truth data for validation and training purposes, a combination of field survey measurements, high-resolution Planet Scope imagery, and Google Earth data was employed. Field survey measurements were conducted using Global Positioning System (GPS) instruments. These measurements involved collecting accurate location information at specific land cover within the study area, which was later used as ground truth data for accuracy assessment. High-resolution Planet Scope imagery, with a spatial resolution of 3 m, was acquired to obtain detailed information about the land cover types. This imagery was also used as a reference dataset for training the machine learning algorithms and validating the classification results. In addition to GPS data and Planate Scope, Google Earth data, containing updated satellite imagery and land cover information, was also utilized as a reference dataset for cross-validation. A total of 5764 ground GPS items and 1718 items from PlanetScope, together with Google Earth map points, were randomly collected for the main agricultural season (September 2020–February 2021) in the study area. This additional dataset helped to strengthen the confidence and accuracy of the classification results.

### 2.3. Image Processing

### 2.3.1. Band Selection

The first steps in image classification are the selection and resampling of the bands. This was necessary to ensure consistency and compatibility between different bands during the subsequent analyses. Based on the spectral characteristics of the study area and target features, appropriate bands from the Sentinel-2 imagery were selected for analysis. This study used 9 spectral bands, such as three visible bands and NIR, as well as four vegetation red-edge bands and two SWIR bands. To ensure spatial uniformity, red-edge (B5, B6, and B7) and SWIR (B11 and B12) were resampled from 20 m to 10 m spatial resolution by using nearest neighbor interpolation to align them with the 10 m spatial resolution bands [92–95]. The selection aimed to optimize the differentiation of the desired land cover classes.

Furthermore, the study area experiences heavy rain during the Meher season (cropping) season, in which cloud coverage poses a challenge for optical remote sensing data. As a result, it is crucial to apply cloud-masking techniques to mitigate the impact of cloud coverage in optical imagery [96,97]. However, cloud masking may result in information

loss, particularly in agricultural areas where cloud cover can be persistent during the rainy season [98,99]. Therefore, by incorporating the Sentinel-1 radar images with the cloud-masked Sentinel-2 optical images, the radar data can compensate for the missing information caused by cloud coverage in the optical images [100–102].

### 2.3.2. Spectral Indices for LULC Detection

Various spectral indices were calculated from the selected optical bands to identify specific land cover characteristics and improve classification accuracy. In this study, the 10 commonly used indices include Normalized Difference Vegetation Index (NDVI) Equation (1), Enhanced Vegetation Index (EVI) Equation (2), Green Normalized Difference Vegetation Index (GNDVI) Equation (3), Bare Soil Index (BSI) Equation (4), tasseled cap wetness index (TCW) Equation (8), Tasseled cap greenness index (TCG) Equation (7), Modified Normalized Difference Water Index (MNDWI) Equation (6), and Normalized Difference Water Index (NDWI) Equation (5), were derived from the selected bands to enhance the LULC classification. In addition to indices from Sentinel-2, the most commonly used radar-derived index, such as the radar ratio index (VV/VH) Equation (9) and modified radar vegetation index (mRVI) Equation (10) [103,104], was developed from Sentinel-1 (Table 1).

**Table 1.** Summary of spectral indices.

| Index | Formula | Reference | Equation |
|---|---|---|---|
| NDVI | $\text{B8} - \text{B4}/\text{B8} + \text{B4}$ | [105] | (1) |
| EVI | $2.5 \times ((\text{B8} - \text{B4})/(\text{B8} + 6 \times \text{B4} - 7.5 \times \text{B2} + 1))$ | [106,107] | (2) |
| GNDVI | $(\text{B8} - \text{B3})/(\text{B8} + \text{B3})$ | [108,109] | (3) |
| BSI | $(\text{B11} + \text{B4}) - (\text{B8} + \text{B2})/((\text{B11} + \text{B4}) - (\text{B8} + \text{B2}))$ | [110,111] | (4) |
| NDWI | $(\text{B3} - \text{B8})/(\text{B3} + \text{B8})$ | [112] | (5) |
| MNDWI | $(\text{B3} - \text{B11})/(\text{B3} + \text{B11})$ | [113,114] | (6) |
| TCG | $(-0.28481 \times \text{B2} - 0.24353 \times \text{B3} - 0.54364 \times \text{B4} + 0.72438 \times \text{B8}$ $+0.084011 \times \text{B11} - 0.180012 \times \text{B12})$ | [115] | (7) |
| TCW | $(0.1509 \times \text{B2} + 0.1973 \times \text{B3} + 0.3279 \times \text{B4} + 0.3406 \times \text{B08}$ $-0.7112 \times \text{B11} - 0.4572 \times \text{B12})$ | [115] | (8) |
| Ratio | $\text{VV}/\text{VH}$ | [103] | (9) |
| mRVI | $\frac{(\text{VV})}{((\text{VV}+\text{VH}))^{0.5}}(4\text{VH})/(\text{VV} + \text{VH})$ | [116] | (10) |

where B2, B3, and B4 are visible bands, B5, B6, and B7 are red-edge bands, B8 is NIR band, and B11 and B12 are shortwave infrared (SWIR).

### 2.4. Classification

Machine learning algorithms have proven to be effective in remote sensing applications, including land cover classification and mapping. Machine learning algorithms can learn patterns and relationships in the satellite data based on various features, such as spectral signatures, indices, and contextual information, allowing for the classification of different land cover types in the study area [117–119]. To perform LULC classification, the reflectance and polarization data with two images (Sentinel-2 and 1) bands were acquired for the Meher crop season between September 2020 and February 2021. In this research, we employed a random and stratified sampling technique to collect ground truth data for classification and adjusted agricultural land area estimation. To ensure an effective assessment of the classifier performance, the reference data were subdivided into two sets: 70% for training the classifiers and the remaining 30% for testing.

Four commonly used machine learning algorithms—RF, SVM, CART, and GTB [19,66,120–123]—were employed for the classification of the study area. The classification processes were carried out using both Sentinel-2 data alone and the integration of Sentinel-2 and Sentinel-1 data. To perform LULC classification, image pre-processing

was performed to obtain reflectance and polarization data with two image bands that were acquired for the Meher crop season between September 2020 and February 2021. The training dataset, consisting of labeled pixels from high-resolution reference images and ground truth data, along with the selected spectral bands and indices, was used to train the machine learning models. The classifiers were trained to differentiate the five defined land cover classes based on their spectral signatures. The trained models were then applied to the entire study area to classify the unknown pixels into the respective land cover classes. This process involved assigning a predicted class label to each pixel based on its spectral characteristics and proximity to the labeled training pixels. In this study, the following classifiers were used for LULC classification:

Random Forest: RF is a widely used algorithm for land cover classification using remote sensing data due to its ability to handle outliers, perform well with high-dimensional datasets, achieve higher accuracy than other classifiers, and increase processing speed by selecting important variables. It is a decision tree-based ensemble learning method that combines a big ensemble regression and classification tree method [124]. The classification and prediction performance of the random forest classification model depends on the optimization of the two primary parameters called the number of trees (Ntree) and the number of features (Mtree) [125–127], which makes it more popular than other machine learning algorithms [128–130]. Recently, several studies have demonstrated that the use of RF in the field of remote sensing applications can achieve good results for the classification of LULC [131–134]. It can handle a wide range of data, including satellite imagery and numerical data [64,135].

Gradient Boosting: The concept of gradient boosting was introduced by Friedman [136]. GTB base model is a robust tree-based data mining model that is flexible for processing different types of data, such as continuous and discrete data [117,137]. It involves fitting a parameterized function to pseudo residuals using additive models in a sequential manner. In the case of Gradient Tree Boosting (GTB), a decision tree is employed as the base learner [138]. It maximizes high-order feature information, generalizes without scaling, and representations by iteratively combining weak learner ensembles into stronger ensembles. GTB outperforms other ensemble classifiers in classification accuracy by using negative gradient loss values in each iteration to fit regression tree residuals [139]. GTB has gained attention in LULC mapping due to its ability to handle imbalanced datasets. Its robustness to outliers makes it suitable for LULC mapping in complex landscapes with high overall accuracy [82,135,140].

Support Vector Machine: Support Vector Machines (SVM) is a non-parametric supervised classification algorithm designed to determine an optimal hyperplane for classifying various classes in the feature space [141,142]. It is based on the principle of risk minimization, which maximizes and separates the hyper-plane and data points closest to the hyperplane spectral angle mapper. The algorithm learns to differentiate between various classes by selecting the hyperplane that maximizes the difference between them [143,144]. In the context of LULC, SVM uses labeled training data, with each sample assigned to a distinct land use or land cover category [145].

Although the polynomial and radial basis function (RBF) kernels have been used commonly in remote sensing, RBF is the most commonly employed approach for LULC classification and gives a higher level of precision than the other classic methods. It requires a good kernel function to reliably build hyper-planes and reduce classification errors [146–148].

Classification and Regression Trees: The Classification and Regression Trees (CART) is a multipurpose machine learning algorithm that uses decision tree principles to address both regression and classification problems [149]. It is an accurate image classification technique that provides the advantages of simplicity and fast execution. However, the algorithm experiences overfitting in the decision tree and generates complicated trees [139]. It works by recursively partitioning the training data into smaller subsets using binary splits [150,151]. It recursively partitions the outcome, in this case, the spatial pattern of interest, into progressively homogeneous subgroups, similar to RF, based on information

provided by the predictor variables [152,153]. Data partitioning proceeds in a stage-wise fashion, which means that earlier split values are not taken into account in successive partitions [154].

*2.5. Accuracy Assessments*

The accuracy of the classification results was evaluated by using two primary metrics, including (OA, Equation (11)), and kappa coefficient (Ka, Equation (12)). The OA represents the overall agreement between the predicted and reference datasets, while the kappa coefficient accounts for the agreement beyond what would be expected by chance alone [155]. In addition to the above, the error of omission and commission was calculated [156]. To provide a comprehensive assessment of the classification results and evaluate the class-wise accuracy, other evaluation metrics, such as F-score (Equation (13)) [157] and Figure of Merit (Fm, Equation (14)) [21], user and producer accuracy [158] were calculated. These metrics evaluate the overall quality of the classification, considering both omission and commission errors. Specifically:

$$OA = \frac{\sum_{k=1}^{n} C_{kk}}{n} \tag{11}$$

where $C_{kk}$ is the row and column $k$ value of the confusion matrix cell and n represents the total number of classes on the map.

$$Ka = \frac{N\sum_{\substack{i=1 \\ j=i}}^{n} d_{ij} - \sum_{\substack{i=1 \\ j=1}}^{n} ri * cj}{N^2 - \sum_{\substack{i=1 \\ j=1}}^{n} ri * cj} \tag{12}$$

where $N$ is the number of pixels, $ri$ and $cj$ are the total number of rows and columns in the error matrix, $n$ is the number of classes, and $d_{ij}$ is the diagonal elements of the confusion matrix

$$F\text{-}score = 2 * \frac{PA * UA}{PA + UA} \tag{13}$$

where $PA$ and $UA$ are the producer and user accuracies, respectively;

$$Fm = \frac{OA}{C_e + O_e + OA} \tag{14}$$

where, $C_e$ and $O_e$ are commission and omission errors, respectively, and $OA$ is the overall accuracy.

## 3. Results

*3.1. Land Use Land Cover Maps*

This study examines the potential of freely available Sentinel-1 and Sentinel-2 datasets and, in the case of the Oromia region, evaluates the performance of machine learning algorithms on agricultural mapping and acreage estimation in small and fragmented farmlands. The result describes the classification results using MLAs of RF, SVM, GTB, and CART classifiers performed on Sentinel-2 (optical data) and the integration of Sentinel-2 with Sentinel-1 (microwave) data. The study combined ground truth from GPS field surveys and references from high-resolution PlanetScope and Google Earth images for accuracy assessments of the classifier. A total of 7482 ground sample points were collected, with the following distribution across classes: water (445), agriculture (1873), vegetation (564), bare land (2636), and built-up (1964). The variance in sample sizes is attributed to easy accessibility and the spatial distribution characteristics inherent to each land cover type. In this study, five classes of surface type were identified: Agriculture, Vegetation, Built-up, Bare land, and Water [159,160]. These classes were selected on the basis of the specific physical characteristics of the study area.

The initial land use and land cover (LULC) class was determined by analyzing class responses in the optical image bands. To improve classification accuracies, different spectral

indices have been developed from the original bands of sentinel products. These indices, as stated in Table 1, give supplemental levels of information that are utilized as input for the classification algorithms. Previous research has proved the significance of these indexes for enhancing the accuracy of land use and land cover (LULC) analyses [82,161].

The second rule set is built upon this by incorporating the SAR VV and HV polarizations, along with related indices. After performing separate classifications using the optical approach and the synergistic approach, two thematic maps representing land use and land cover (LULC) were generated. These maps are displayed below, along with the results of the accuracy assessment.

Figure 3 presents the optical image classified maps obtained by the four machine learning algorithms (MLAs) for the Meher crop season in the study area. These LULC maps produced by Sentinel-2 data with RF, SVM, GTB, and CART models are presented in Figure 3a–d, respectively. The analysis of the graphical distribution of the classes in these maps reveals a consistent pattern in the major LULC units, with minor variations observed in the southern and some eastern areas, where the density of bare land is higher. This can be attributed to the region's mountainous terrain and shrubland, which have been inaccessible to the public since the previous regime, as well as the dry climate leading to drought-affected areas in the region. From the map, it is observed that the CART algorithm-based map depicts more urban areas than other MLAs, which are represented by red color.

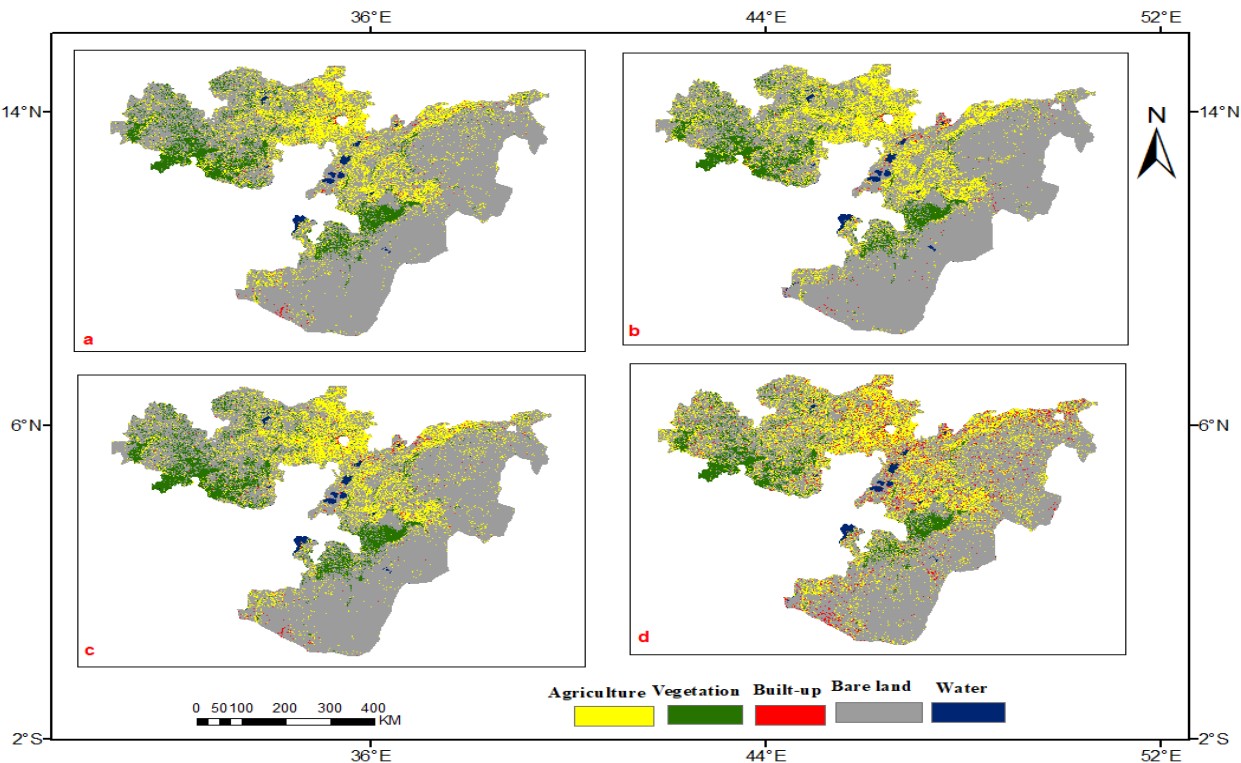

**Figure 3.** Results for the mapping of LULC and agricultural landscapes derived from the classification of Sentinel-2 data using the following: (**a**) RF; (**b**) SVM; (**c**) GTB; (**d**) CART.

Additionally, minor dissimilarities are observed in the central area, where the largest land cover, agriculture, is located. Three classification algorithms, RF, SVM, and GTB, exhibit a consistent pattern in vegetation land cover with insignificant differences. Regarding the water cover, a relatively similar spatial extent is evident in the classification maps generated by all classifiers. The map below shows that the agricultural land cover class spatial pattern and extent were very similar for RF and GTB, Figure 3a,c.

The spatiotemporal availability of earth observation (EO) data, typically associated with vegetation greenness and crop growth in the visible and infrared portion of the electromagnetic spectrum, may be problematic when clouds prevent measuring the land

surface [162,163]. The combination of optical and SAR data allows for a more comprehensive representation of biophysical and structural information on target objects, which in turn improves crop mapping [164,165].

In this study, we assess the performance of SAR and Optical integrated images for agricultural mapping and acreage estimation in small, fragmented farmlands. Therefore, LULC classification maps for S2 and S1 integration were generated using the same four supervised classification techniques. The findings of the study reveal a significant enhancement in the geographical distribution and visual interpretation of classification when utilizing combined imagery, as compared to relying solely on optical-only imagery.

The results show that, among the four algorithms examined, SVM, RF, and GTB demonstrated similar patterns in the homogeneity of class distribution related to Agriculture, vegetation cover, built-up area, water, and bare land. In contrast, the CART displayed marked discrepancies in the generated maps, primarily characterized by an overestimation of the built-up and agricultural classes, suggesting a lower level of precision in its results (Figure 4d).

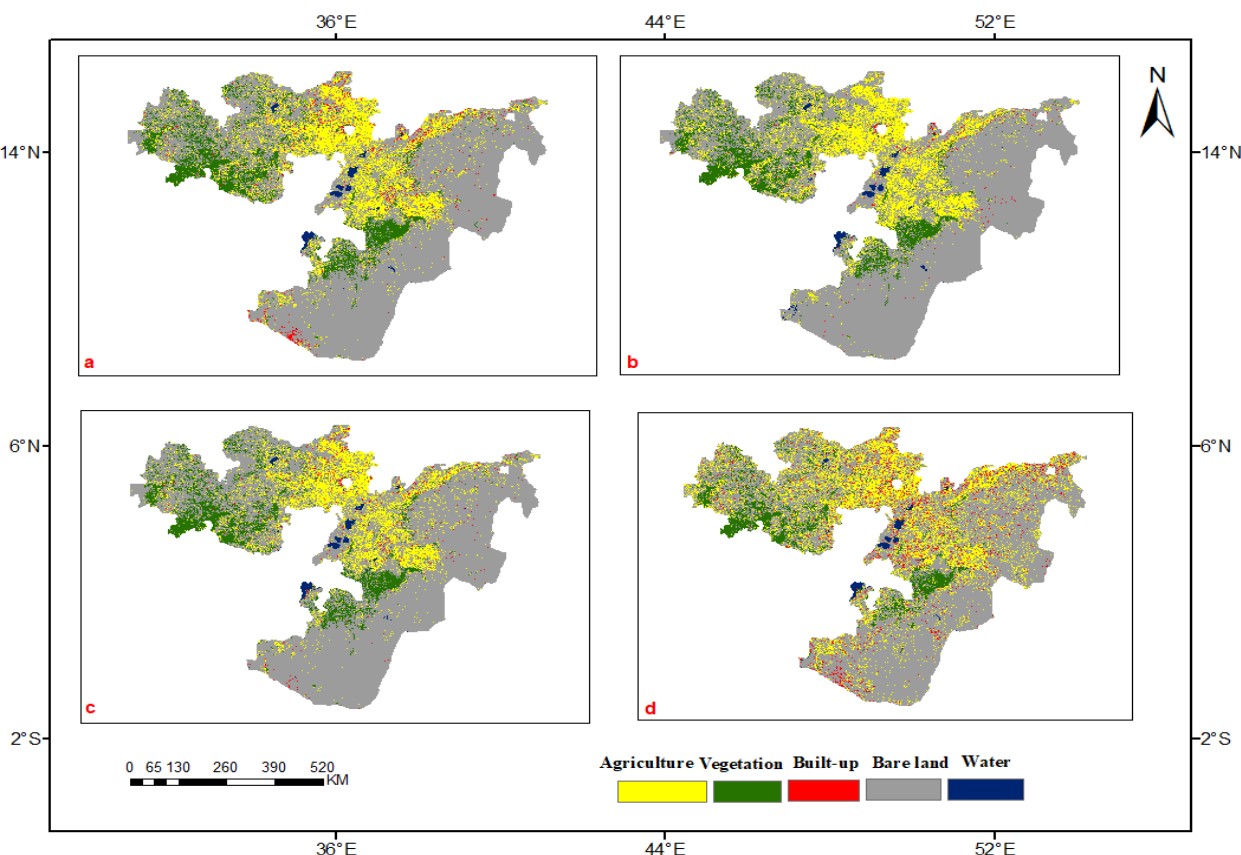

**Figure 4.** SAR-Optical integrated LULC map using different classifiers. The position of the maps is the same as in Figure 3. (a) SVM; (b) RF; (c) GTB; (d) CART.

The visual inspection of LULC revealed that the combined datasets could provide the best classification and were comparable more closely to current land covers based on SVM, RF, and GTB classification algorithms (Figure 4a–c). Most of the areas near the main southern parts and toward the eastern portion of the study area are highly covered with bare lands, also classified in the bare lands category in the classified results. Furthermore, the central and northwestern parts of the region are highly covered by agriculture and forest, respectively, and are also classified as agricultural and vegetation land cover. The map shows the types of LULC categories that exist in the area, with cultivated (Meher season) areas dominating the landscape next to the bare land. In contrast, water and settlements

constitute a smaller percentage. The maps depict the many LULC classifications that exist in the area.

### 3.2. Quantitative Evaluation

The accuracy assessment of the classification was conducted in two approaches, directly by using the pixel classification and by using the class as strata to calculate unbiased class-wise accuracies, using various metrics such as overall accuracy, kappa coefficient, f-score, and figure of merit. In order to enhance classification accuracies, additional spectral indices are derived from the original bands of sentinel products [82,161]. The OA and Ka values, as well as the F-score and Fm measures, were computed from the confusion matrixes for RF, SVM, GTB, and CART in Tables 2 and 3, respectively.

**Table 2.** Optical map accuracy metrics.

|  |  | **UA** | **PA** | **F-Score** | **FM** |
|---|---|---|---|---|---|
|  | Agriculture | 0.88 | 0.76 | 0.81 | 0.71 |
|  | Vegetation | 1.00 | 0.98 | 0.99 | 0.98 |
| SVM | Built-up | 0.96 | 0.98 | 0.97 | 0.94 |
|  | Bare land | 0.86 | 0.92 | 0.89 | 0.80 |
|  | Water | 0.95 | 0.92 | 0.94 | 0.87 |
|  | Agriculture | 0.83 | 0.79 | 0.81 | 0.70 |
|  | Vegetation | 1.00 | 0.98 | 0.99 | 0.98 |
| RF | Built-up | 0.93 | 0.91 | 0.92 | 0.85 |
|  | Bare land | 0.84 | 0.88 | 0.86 | 0.75 |
|  | Water | 0.97 | 0.91 | 0.94 | 0.88 |
|  | Agriculture | 0.83 | 0.80 | 0.82 | 0.71 |
|  | Vegetation | 0.99 | 0.98 | 0.98 | 0.97 |
| GTB | Built-up | 0.93 | 0.92 | 0.93 | 0.86 |
|  | Bare land | 0.85 | 0.89 | 0.87 | 0.77 |
|  | Water | 0.97 | 0.90 | 0.93 | 0.87 |
|  | Agriculture | 0.75 | 0.78 | 0.77 | 0.64 |
|  | Vegetation | 0.99 | 0.98 | 0.98 | 0.96 |
| CART | Built-up | 0.86 | 0.89 | 0.88 | 0.77 |
|  | Bare land | 0.83 | 0.81 | 0.82 | 0.70 |
|  | Water | 0.95 | 0.88 | 0.92 | 0.83 |

**Table 3.** Joint SAR/optical data maps accuracy summary.

| **Classifier** | **Class** | **UA** | **PA** | **F-Score** | **FM** |
|---|---|---|---|---|---|
|  | Agriculture | 0.92 | 0.87 | 0.89 | 0.82 |
|  | Vegetation | 0.99 | 0.98 | 0.98 | 0.97 |
| SVM | Built-up | 0.99 | 0.98 | 0.98 | 0.97 |
|  | Bare land | 0.91 | 0.95 | 0.93 | 0.87 |
|  | Water | 0.93 | 0.91 | 0.92 | 0.86 |
|  | Agriculture | 0.91 | 0.89 | 0.89 | 0.82 |
|  | Vegetation | 0.99 | 0.99 | 0.99 | 0.97 |
| RF | Built-up | 0.96 | 0.94 | 0.95 | 0.90 |
|  | Bare land | 0.88 | 0.91 | 0.89 | 0.81 |
|  | Water | 0.98 | 0.94 | 0.97 | 0.92 |
|  | Agriculture | 0.89 | 0.87 | 0.88 | 0.80 |
|  | Vegetation | 0.99 | 0.98 | 0.98 | 0.97 |
| GTB | Built-up | 0.96 | 0.94 | 0.94 | 0.90 |
|  | Bare land | 0.89 | 0.91 | 0.90 | 0.82 |
|  | Water | 0.82 | 0.87 | 0.92 | 0.75 |
|  | Agriculture | 0.83 | 0.86 | 0.84 | 0.74 |
|  | Vegetation | 1.00 | 0.96 | 0.98 | 0.96 |
| CART | Built-up | 0.95 | 0.91 | 0.93 | 0.86 |
|  | Bare land | 0.83 | 0.85 | 0.84 | 0.73 |
|  | Water | 0.98 | 0.88 | 0.93 | 0.86 |

Despite the complex nature of the landscape and fragmented small plot sizes within the study area, which were classified into five distinct land use land cover (LULC) categories, the LULC maps produced by the MLAs showed high-accuracy assessment results. Figure 5 illustrates the comparative analysis of OA, Kapa, and F-Score achieved by all MLAs for LULC classification during the Meher season. For the Sentinel-2-only data, the selected algorithms exhibited notable performances, with the Support Vector Machine (SVM) classifier achieving the highest OA value of 89.9%, while the Classification and Regression Tree (CART) classifier attained the lowest OA value (83.4%) (Figure 5).

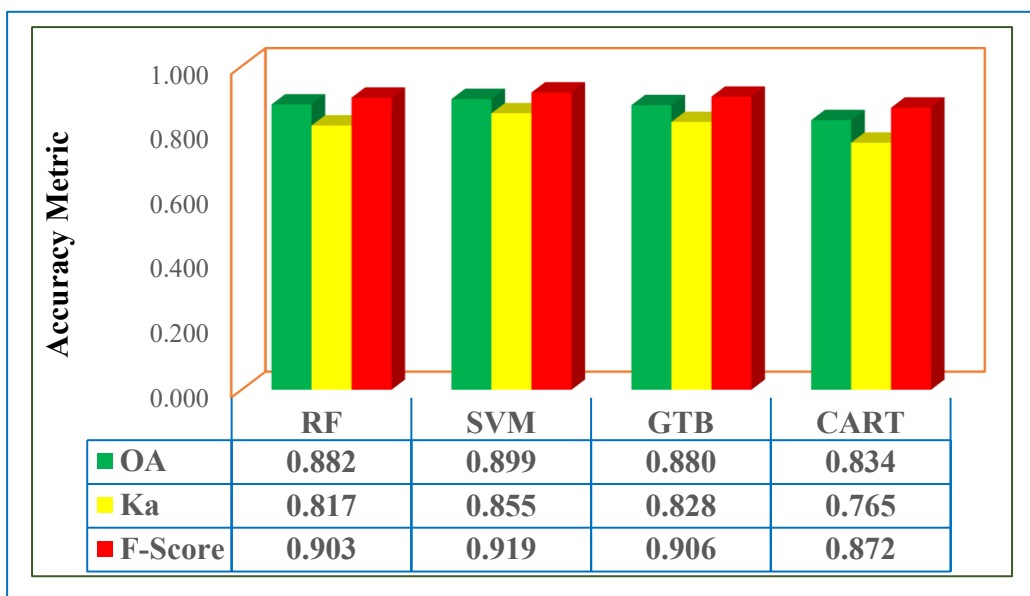

| | RF | SVM | GTB | CART |
|---|---|---|---|---|
| ■ OA | 0.882 | 0.899 | 0.880 | 0.834 |
| ■ Ka | 0.817 | 0.855 | 0.828 | 0.765 |
| ■ F-Score | 0.903 | 0.919 | 0.906 | 0.872 |

**Figure 5.** Summary of the accuracy metrics for MLAs applied to optical data.

In the case of the RF and GTB classifier, employing optical-only imagery results in an average accuracy of 88%, while the CART classifier achieves an average accuracy of 83%.

In addition to overall accuracy, the accuracy assessment was carried out, which exhibits users (UA) and producer's accuracy (PA) of each class of study area generated through SVM, RF, GTB, and CART (Table 2). The agricultural class of the LULC map generated by the SVM yields the highest UA values of 88%, while the CART yields less UA and values of 75%. The findings of the assessments also indicate the UA and PA of agricultural land use retrieved by RF and GTB are equal, measuring 83% and 80%, respectively. According to the results, SVM has the highest UA as compared to another classifier but less PA than RF and GTB. From Table 2, the accuracy metrics f-score for agricultural land cover is greater than 0.8 for RF, SVM, and GT, while it gave a lower result with the CART methods, which is 0.77. This indicates the potential of SVM, RF, and GTB for identifying and mapping agricultural areas within small and fragmented farmlands. The figure of merit for vegetation was over 95% for all classification algorithms.

There is variation in the accuracy of MLAs; this was confirmed by the previous study that demonstrates the highest achieved accuracies for each classifier, which vary depending on the type of imagery, input dataset, and training data configuration [135,166,167]. Furthermore, according to previous studies, any accuracies exceeding 85% are thresholds considered satisfactory for land use and land cover (LULC) applications [168–170]. Therefore, the findings of our analyses revealed that the obtained result is satisfactory for mapping and monitoring agricultural land in small and fragmented land farming systems in developing countries.

Additionally, the objective of this study was to determine the impact of S1-VH and S1-VV satellite images on enhancing the accuracy of classification in an area having small and fragmented farmlands. In the same ways, classification models, RF, SVM, GTB, and

CART, were implemented on the integrated S1-VH and S1-VV polarization data along with the S2 satellite.

As compared to optical images alone, when SAR and optical imagery are combined, the accuracy of SVM increases to an average of 94%, while RF and GTB improve to an average of 92% and 91%, respectively (Figure 6). Similarly, the Classification and Regression Trees (CART) classifier also demonstrates the highest accuracy of 87% when utilizing the combined imagery.

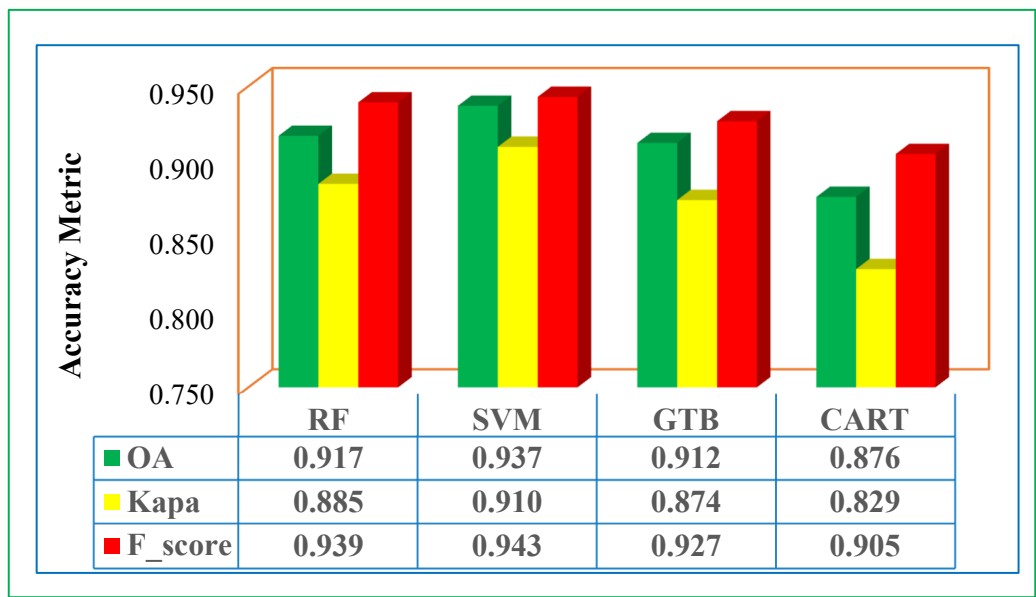

| | RF | SVM | GTB | CART |
|---|---|---|---|---|
| ■ OA | 0.917 | 0.937 | 0.912 | 0.876 |
| ■ Kapa | 0.885 | 0.910 | 0.874 | 0.829 |
| ■ F_score | 0.939 | 0.943 | 0.927 | 0.905 |

**Figure 6.** Result of the accuracy metric for MLAs applied to SAR and optical.

Therefore, the synergistic imagery achieved the highest level of accuracy, indicating the incorporation of indices derived from Sentinel-1 images further improved the model performance marginally. Furthermore, the feature importance analysis highlighted the substantial contribution of radar data to the classification process. Previous studies prove that the addition of arithmetic operations of vertical and horizontal polarization as input for classification yields an increase in the accuracy of LULC classification approaches [81,171,172].

Furthermore, the classification accuracies at a class-wise level were evaluated using the validation dataset. Table 3 illustrates the class-wise accuracies (UA and PA) achieved by the SVM, RF, GTB, and CART models. These accuracies offer important perspectives into the specific contributions of each model in the detection and classification of various land cover in the study area, as indicated by the producer's and user's accuracies.

When compared with other classes, vegetation, built-up, and water bodies performed well, with more than 90% user and producer accuracy for the SVM, RF, and GTB models, while the CART gave UA and PA of less than 90% for the built-up class. Among the four classifiers tested, SVM and RF demonstrated the highest UA and PA values across all classes. While in both methodologies, SVM outperformed the other classifiers in terms of producer and user accuracy. SVM achieved a PA and UA exceeding 90% for all classes, while RF achieved a PA and UA exceeding 90% for all classes except for the bare land class, which had a UA of 88%. On the other hand, the GTB and CART classifiers showed higher accuracy (greater than 90%) for the vegetation and built-up area classes but had lower UA and PA for the other classes compared to SVM and RF. These results suggest that SVM and RF outperformed the other classifiers and are more suitable for agricultural mapping and estimating acreage in small and fragmented farmlands. The findings of the study confirmed similar studies that observe satisfactory levels of classification accuracy through the synergistic utilization of S2 multi-spectral and S1 polarization data [81,172].

Moreover, when comparing the accuracy between standalone Sentinel-2 classification and integrated Sentinel-2/Sentinel-1 classification, the results revealed that the integrated classification approach yielded higher accuracy. This improvement could be attributed to the complementary information provided by combining the optical and radar data sources, resulting in more robust and accurate classification results [47,173,174].

### 3.3. Acreage Estimation and Implications for Small Farm Holdings

The accurate estimation of acreage in small farm holdings has an important role in agricultural planning and management. The estimated areas for each LULC class provide valuable insights into the distribution of various crops within the study area. In this study, the area measurements were obtained for five land use land cover categories. The figure below illustrates the proportional distribution of land use land cover extracted from LULC maps generated through selected machine learning algorithms for classifications for optical image only (Figures 7 and 8) and the combination of optical and radar (Figures 9 and 10).

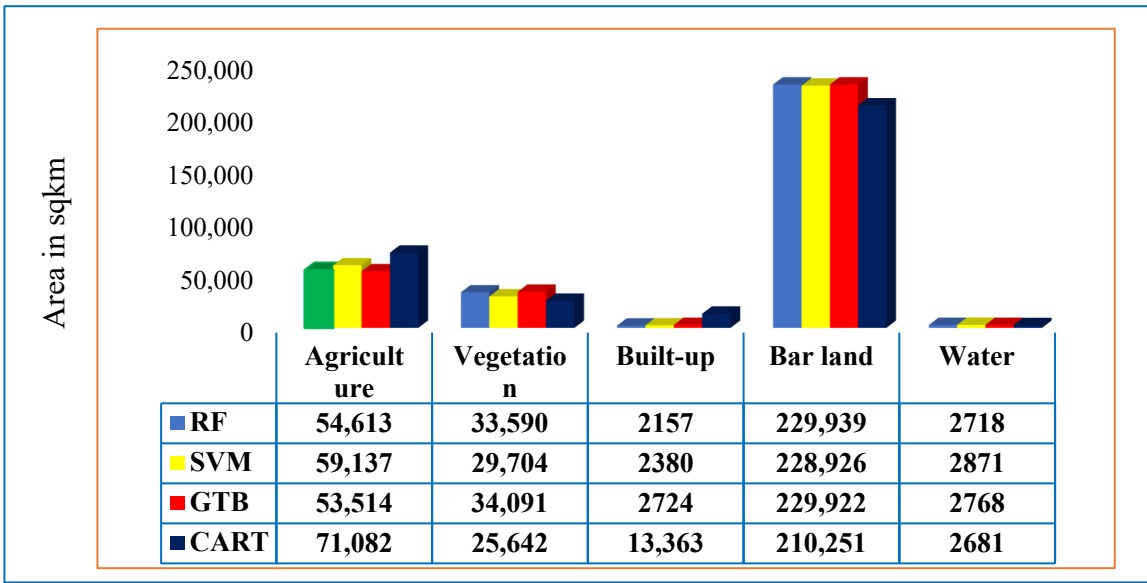

Figure 7. Areas of LULC for different MLAs based on Sentinel-2.

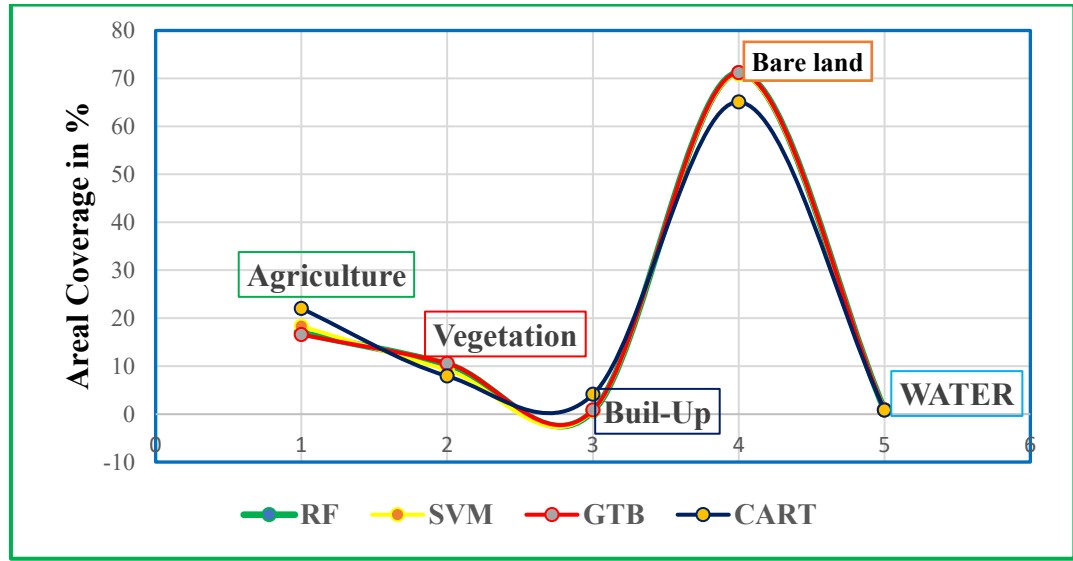

**Figure 8.** Comparative summary of the area in % for MLAs applied to optical data.

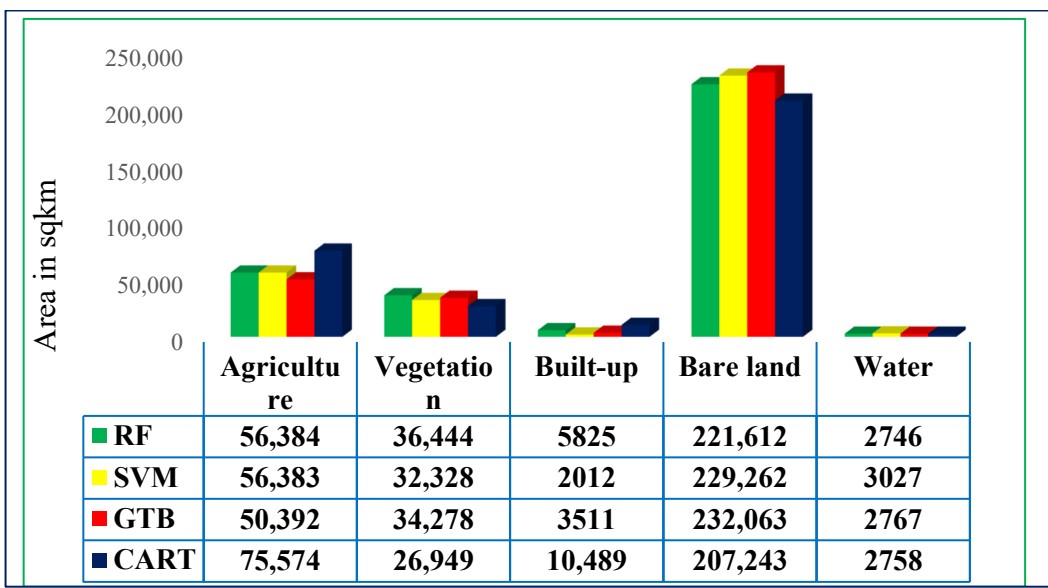

**Figure 9.** Areas of LULC for different MLAs based on Sentinel-1 and 2.

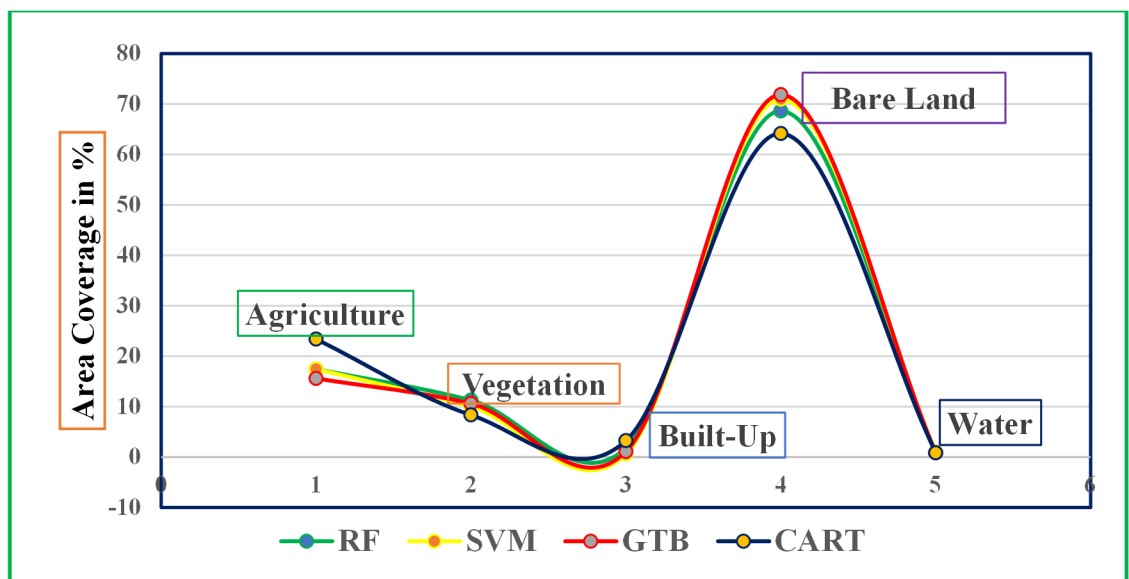

**Figure 10.** Comparative summary of the area in % for MLAs applied to SAR and optical data.

The Sentinel-2 classified image area results (Figure 7) showed that the area obtained for water classes was very close and similar across all four algorithms, indicating the robustness of the classification for this specific class. Similarly, the three algorithms, SVM, RF, and GTB, yielded fairly similar area measurements for agricultural land, vegetation, and bare land classes. Among these three algorithms, RF and GTB provided the very closest results.

From the findings, we observe that the RF and GTB algorithms achieved the same coverage for agricultural land, measuring 16.5%, when applied to optical imagery, while the CART algorithm exhibited the highest coverage for this class, about 22% of the total area (Figure 8). Conversely, the SVM, RF, and GTB classifiers demonstrated nearly similar coverage for vegetation accounts, 9.2%, 10.4%, and 10.5%, respectively, with the CART classifier having the least coverage, which measured 7.9%. Settlement coverage exhibited an inverse relationship with vegetation cover, with the CART classifier achieving the highest coverage (4.14%), followed by GTB (0.84%) and SVM (0.74%) classifiers, and the RF

classifier exhibiting the lowest coverage measuring 0.64%. Notably, all classifiers achieved similar levels of coverage for water surface coverage (Figure 8).

By leveraging the complementary strengths of radar and optical data, acreage estimation becomes more robust, accounting for variations in crop phenology, crop structure, and soil conditions. To assess the spatial extent of different land use and land cover (LULC) classes, the areas of the S2-S1 maps of Figure 4 were generated. The results, presented in Figure 9 and the associated Figure 10, demonstrate significant variation in the areas of LULC classes across the models. Specifically, similar to Sentinel-2 alone classification, the CART model tends to overestimate agricultural land cover, measuring 75,574 ha (23.39%) of the total area, and built-up area, measuring 10,489 ha (3.24%) compared to the other models. On the other hand, it underestimates bare land and vegetation areas, measuring 207,243 ha (64.15%) and 26,949 ha (8.34%) in comparison to the other models. Interestingly, the SVM and RF models provide an equal estimate for agricultural land, measuring 17.45% of the total area, compared to the other three models.

Interestingly, the area of the agricultural land class by RF and SVM shows a strong correlation with the area of agriculture obtained from the central statistics data office, which is measured as 5,501,300.583 ha.

According to our findings, there are significant discrepancies in the acreage estimation obtained from the classification of Sentinel-2 alone (Figure 8) and the synergy of Sentinel-2 and Sentinel-1 data (Figure 10) for all land use and land cover classes. Among the four algorithms tested, the Support Vector Machine classifier demonstrated the smallest differences in acreage estimation between the two methods. The SVM algorithm displayed differences of less than 1% for each land use and land cover class between the two methods. This suggests that SVM are highly accurate and reliable in estimating acreage for agricultural land in small and fragmented farmlands.

Following SVM, the Gradient Boosting algorithm exhibited the second smallest variations in acreage estimation results. On the other hand, the Random Forest and Classification and Regression Tree classifiers displayed differences of up to 2% for certain land use and land cover categories. These findings highlight the effectiveness and robustness of the SVM classifier in identifying and mapping agricultural land in small and fragmented farmlands compared to the other classifiers utilized in this study.

However, the above approaches for area estimation are based on the summation of the area covered by the land cover map classification pixel, which does not adjust for classification errors in the map caused by class confusion [47,173,175]. Therefore, by incorporating the known area proportions of the map class in the stratified estimators of the overall and class-wise accuracies, the uncertainty and confidence interval in all classes (strata) can be computed.

Table 4 shows that the user's accuracy of agricultural land mapping using Sentinel-2 alone varies between 75 and 88% depending on the classifier used. Conversely, when integrating Sentinel-2 and Sentinel-1 data, the user's accuracy was between 83% and 92%. However, it is noted that the producer's accuracy for agriculture was relatively low, ranging from 65 to 77%. This fact demonstrates that the maps failed to capture some portions of the agricultural area in the reference data. From the findings of this study, a high user accuracy coupled with a low producer accuracy for agricultural land implies that the majority of the land labeled as agricultural in the LULC map reflects the land cover for agriculture in the reference data.

As depicted in Table 4, considering the uncertainty in area estimation associated with pixel counting at the 95% confidence level, the true area of agricultural land may vary between 60,662 and 81,658 km$^2$, covering 19–25% of the total surface in the study region.

To validate the accuracy of the results, we compared the area of the agricultural LULC class derived from each machine learning classifier in both classification results with data from Ethiopia statistics service as well as from ESA and ESRI Global land cover databases [176–178]. The results showed that SVM, RF, and GTB were found to be very close to the crop area data obtained from the country statistics data center. This indicates that

these machine learning algorithms successfully captured and classified agricultural land use with high precision and accuracy. To further explain this, let us take a sample number.

**Table 4.** Summary of unbiased class-wise accuracies and area estimation.

| Class | Dataset | MLAs | UA | PA | Area in km | ±95% CI | OA |
|-------|---------|------|------|------|------------|---------|-----|
| Agriculture | S2_S1 | SVM | 0.92 | 0.77 | 67,801.48 | 4119.42 | 92 |
| | | RF | 0.91 | 0.74 | 68,705.93 | 4942.889 | 90 |
| | | GTB | 0.89 | 0.74 | 60,662.78 | 4147.145 | 90 |
| | | CART | 0.83 | 0.76 | 81,658.95 | 5507.65 | 85 |
| | S2 | SVM | 0.88 | 0.65 | 79,785.45 | 5281.748 | 88 |
| | | RF | 0.83 | 0.65 | 70,127.21 | 5124.638 | 86 |
| | | GTB | 0.83 | 0.66 | 68,027.59 | 4991.566 | 86 |
| | | CART | 0.75 | 0.69 | 76,942.89 | 5342.114 | 83 |

The area of agricultural land derived from the country statistics data center is 16.96 percent of hectares. The machine learning classifiers, SVM, RF, and GTB, classified the agricultural land as approximately 17 percent of the total land coverage. This indicates a close match between the classifier results and the ground truth data. Furthermore, the area of the water surface from Global LULC is very similar to the area obtained in this study.

Furthermore, the area of agricultural land derived from machine learning algorithms was compared with agricultural land from the global Land Use and Land Cover (LULC) cover datasets of ESA and ESRI (Figure 11). To address this, the global LULC classifications were reclassified into five classes that were deemed to be highly important and relevant during the study period.

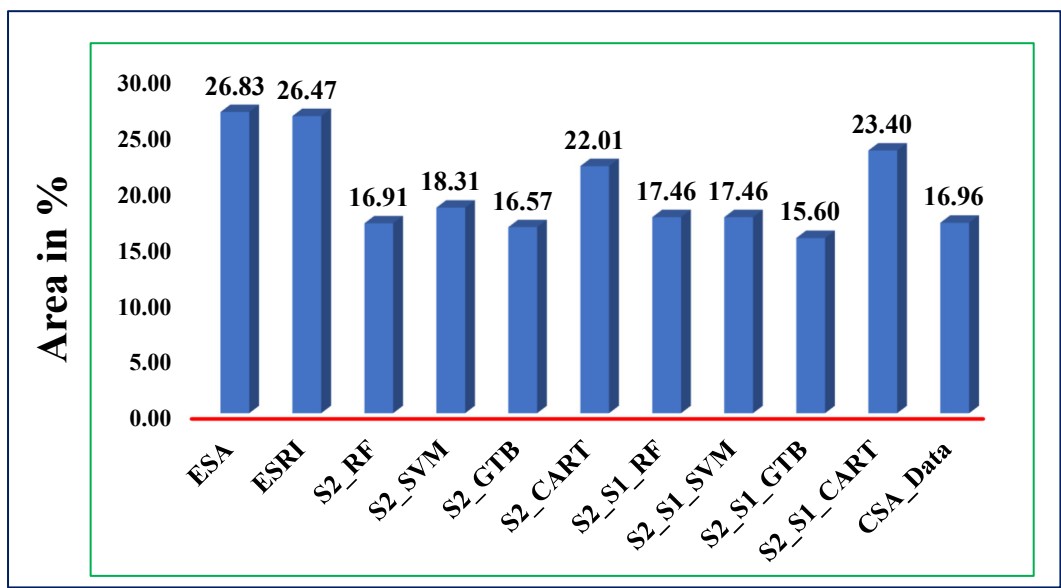

**Figure 11.** Agricultural area from different sources and/or with different methodologies.

The result indicates that the area of agricultural land obtained from the global LULC datasets of ESA and ESRI is 26.5% and exhibited significant differences when compared to the results obtained using the machine learning algorithms, which involve 15–23% of the total coverage at the study area. These differences indicate variations in the classification accuracy and identification of agricultural land between the global datasets and the approach employed in this study. However, it was observed that the land use classes of built-up areas and water surfaces derived from the global LULC datasets demonstrated a closer agreement with the land cover classes obtained through the machine learning

algorithms. This suggests that the classification of built-up areas and water surfaces in the global datasets aligns more closely with the machine learning results at a local level.

The findings also suggest that CART may not be the most suitable algorithm for the accurate estimation of agricultural areas in small farm areas. The overestimation observed in the CART results could be attributed to the complex nature of agricultural land and the limitations of the algorithms in handling such complexities.

## 4. Discussion

The main goals of this study were to assess the application of freely available high-resolution satellite imageries, particularly sentinel products, as well as the performance of the MLAs for detecting and differentiating agricultural areas at small and fragmented farmlands in developing countries.

Agricultural monitoring in sub-Saharan Africa is challenging due to inaccurate agricultural statistics and coarse data analysis [179,180]. However, the free availability of 10 m Sentinel-2 data and an advanced processing platform allows for efficient processing of high spatial resolution data, making crop area maps feasible [7,181,182]. The high spatial resolution satellite allows for fewer mixed pixels in these smallholder agricultural landscapes, resulting in mosaics of fields that are often heterogeneously mixed at lower resolution satellite data [183–185]. In this study, the high-resolution freely available sentinel data were evaluated, and the performances of commonly used MLAs were tested concerning the small and fragmented farmlands.

The classification based solely on Sentinel-2 imagery yielded reasonably accurate results, with overall accuracies ranging from 83.4% to 89.9% for the different machine learning algorithms utilized. The obtained accuracy metrics showed that all MLAs had highly acceptable accuracy values, with SVM having the highest values for overall accuracy and kappa coefficients. The overall accuracy of SVM in mapping agricultural land from the other four land covers consisted of an OA of 89.9% and kappa of 0.86. As a result, the agricultural and non-agricultural land cover could be distinguished. The result also indicates that most of the classes exceeded the 90% F-score criteria, with the class relating to vegetation cover achieving a value of 99%. In contrast, the CART-based classification for optical images of agricultural land cover resulted in the lowest F-score value (77%) due to incorrectly classifying pixels from the barren land class as agricultural and built-up. This omission error in the agriculture class is demonstrated by a user accuracy of 75% and a producer accuracy of 78% in the same class.

In this study, although the CART, in some cases, makes errors of omission by misclassifying water as barren land areas, bare land as built-up areas, and bare land as agriculture, the remaining classifiers demonstrate a relatively high level of confidence in differentiating different surface cover of the regions. In particular, RF and SVM delineate water with respective consumer and producer accuracies of 90%, 94%, and 93%, 98%, respectively.

The same study conducted by Ouma et al. [139] compares four machine learning algorithms, RF, GTB, SVM, and multilayer perceptron neural networks (MLP-ANN), for the classification of LULC and they found that SVM, RF, and GTB perform similarly to our findings. Furthermore, the finding of this study was aligned with the study conducted by Basheer [159], who evaluated the performances of SVM. Regarding RF, CART, and ML classification, they found that SVM outperforms the other approaches with an OA accuracy of 89%-94, depending on the data used. Similar to this study, they demonstrate the existence of a very slight difference between GTB and RF in the LCLU classification study using high-resolution data from RapidEye [186].

However, relying solely on optical imagery has limitations in accurately identifying land cover classes, particularly in areas with dense vegetation or cloud cover. This could explain the observed variations in classification accuracy among the different algorithms employed. Several recent studies have acknowledged the challenges associated with the misclassification of land cover classes when using only optical data. The combination of

SAR and multispectral data offers a more comprehensive approach to land cover mapping and can lead to improved results [187].

Therefore, in this study, integrating Sentinel-1 radar data with Sentinel-2 optical data, a significant improvement in classification accuracy was achieved across all machine learning algorithms applied. The overall accuracies ranged from 87.6% to 93.7%, indicating the significance of combining radar and optical data sources for accurate agricultural mapping in small farmlands. The result shows that SVM yields the highest OA at 94% and Ka at 91%, respectively (Table 3). Similarly, different studies demonstrate the highest OA accuracy of classification by integration of optical and radar datasets [188,189].

The results of the study indicate that the integration of Sentinel-1 and Sentinel-2 satellites resulted in an improvement in the overall accuracy of the classification by approximately 3–4.5%. This finding was confirmed by the studies on the improvement in accuracy of the classification by the integration of SAR to optical by >2.5% compared to that obtained using only optical data [175,190]. Furthermore, the previous study has shown that combining Synthetic Aperture Radar (SAR)-optical data improves the classification accuracy of Machine Learning Algorithms (MLAs) by 4%. This improvement was especially noticeable in areas with diverse landscapes and weather conditions that make remote sensing data collection difficult. Similarly, the investigation by Khan et al. [170] confirmed that the addition of VV and VH into Sentinel-2 data increased the kappa coefficient from 75% to 82%.

Our result is aligned with the study that demonstrates that adding the radar-derived index into optical bands increases the accuracy of classification [191]. Furthermore, the Studies conducted by Nicolau et al. [192] and De Luca et al. [175] emphasize the complementary nature of SAR and optical imagery, suggesting that their integration can provide a more comprehensive understanding of land cover characteristics.

The finding indicates that good overall accuracy was achieved by the integrated optical and SAR datasets, represented by a mean F-score equal to 94.28%, which is in line with the outcomes obtained from other studies where the combination of optical and SAR data was used.

This result was confirmed with the study conducted to evaluate the performances of SVM, RF, and K-NN. They found that the SVM technique resulted in the highest OA (88.75) and ka (0.86) to classify LULC by radar and optical integrated dataset and they concluded that the overall accuracy of the integrated dataset is higher than the single image [193]. According to recent reviews, support vector machines (SVM) and random forest (RF) are the most popular machine learning algorithms for classification, with comparable high accuracy [66,146,194], but the conclusion on which classifier performs better in LULC classification is unclear. According to the findings, some classes are wrongly identified, and it was difficult to differentiate between built-up, barren land, and vegetation classes by optical scene compared to the synergistic of optical with radar sceneries due to mixed pixels in a very small plot.

Crop acreage is one of the most important pieces of information needed to quantify food production at the regional or country level, which will be used for the implementation of sustainable agricultural management systems and monitoring the progress toward the SDGs [195]. The findings of this study highlight the importance of carefully selecting the appropriate machine learning algorithm for accurate estimation of area. There is variability in the classification of land use and land cover (LULC) classes between different classifiers [131], as demonstrated in Figures 8 and 10. This discrepancy in LULC classes can be attributed to variations in parameter optimization within the algorithms employed [159]. Furthermore, the differences in area estimation among the classifiers may be attributed to the inherent characteristics and biases of each algorithm and the size of the parcel. Various studies have found that the areas of different land use and land cover (LULC) classes vary depending on the classification technique [121,160,196]. This study also observed variations in the results of four classifiers, where the area under each LULC class of one classifier did not exactly match the area under the same class of another classifier.

However, the map provides an area estimate for agricultural land without considering the uncertainty in pixel counts. Subsequently, through the creation of an error matrix, the error-adjusted/unbiased area of agricultural land, along with the confidence interval, was computed. The result provides additional information into area estimation, which significantly deviates from results obtained solely through pixel counting methodologies.

The LULC classification map provides a single area estimate for each land cover class without a confidence interval. In this study, an error matrix was generated from the class strata sample pixel counts. Subsequently, accuracy assessments and confidence intervals were derived. The results, presented in Table 4, show that the overall accuracy metric using this method aligns with those obtained directly from the pixel-based confusion matrix, with SVM having the maximum OA of 92%. However, slight differences are observed in users' accuracy and significant variation in producer accuracy with different machine learning algorithms. These findings are consistent with previous studies that have demonstrated there is more variation in the producer's accuracies than in the user's accuracies [145,197].

In this instance, the mapped area of agricultural land ranges from 15.6% to 23.4% ha for the integrated dataset and from 16.6% to 22% ha for Sentinel-2-based classification. However, the stratified error-adjusted area estimate for agricultural land is approximately 18.7% to 25.2% for the integrated dataset and 21% to 24.6% for optical image only (Table 4). This disparity can be attributed to the error matrix, which indicates that some of the proportion of the area of agricultural land is omitted from the map. These results are in line with other similar studies that have also demonstrated the improved accuracy of area estimation by considering the uncertainty in the error matrix and confidence interval [198–200].

Furthermore, complicating matters further, the research area is situated in a developing country characterized by unmodernized farming systems with small, fragmented farm sizes. The small size of farmlands coupled with the cultivation of different crops on these diminutive plots poses challenges in identifying and delineating agricultural areas [36]. Additionally, the presence of various types of grass used for grazing and boundary marking further hinders the precise determination of agricultural land and its extent. In such cases, remote sensing technology encounters difficulties in performing accurate LULC classification. Another issue faced during the study was the existence of pastureland between crops. Pastureland typically has a different spectral signature compared to crops, and its presence further complicates the classification process. For example, in some areas where pastureland existed between maize fields, the classification algorithm struggled to differentiate between the two land cover types accurately. This led to misclassifications and reduced accuracy in those specific areas.

## 5. Conclusions

The primary objective of this study was to evaluate the suitability of Sentinel-2 and Sentinel-1 products for the analysis of agricultural land within a small and fragmented farm region during the Meher season of the year 2020/2021. The application of freely available multi-source imagery for agricultural mapping in small-scale farmlands is important for many developing countries that face budget constraints in acquiring high-resolution data. The results reveal the potential of freely available sentinel products, especially when integrating Sentinel-1 and Sentinel-2 data for mapping agricultural areas and estimating acreage within small farmlands in developing countries. The integration of Sentinel-1 data with Sentinel-2 data, along with the use of advanced classification algorithms, can significantly improve the accuracy of agricultural mapping and acreage estimation. When the findings from Sentinel-2 alone were compared to the synergy of Sentinel-2 and Sentinel-1 imagery, the synergistic dataset produced the highest OA and Ka accuracy results for agricultural mapping and acreage estimation on small, fragmented farmlands and heterogeneous cropping parcels.

In terms of accuracy comparison, the findings of this analysis consistently demonstrate that SVM, RF, and GTB yield comparable high accuracies. But SVM outperform with a very slight difference in terms of overall accuracy (OA) and kappa coefficient. It is believed that

the acquisition of an accurate agricultural map with an estimate of accuracy of more than 90% has significance for improving further monitoring and analysis of agricultural land in a small and fragmented farmland region.

In terms of acreage estimation, two approaches were employed: direct calculation from pixel values and stratified sampling using LULC map classes as strata. Notably, the integrated Sentinel-2 and Sentinel-1 approach yielded promising results in both methods. The application of stratified sampling for unbiased area estimation, supported by a 95% confidence interval, revealed that the integrated approach outperformed the Sentinel-2-alone classification, producing results closely comparable to the ground truth data.

These outcomes demonstrate the remarkable potential of freely accessible multi-source remotely sensed data in agricultural mapping and acreage estimation operations in small farm holdings. They also further demonstrate the significant capability of such data in supporting the monitoring and management of agricultural resources in small-scale farmlands within developing countries.

**Author Contributions:** Conceptualization: T.E.M.; Methodology: T.E.M. and P.G.; Formal analysis and investigation: T.E.M., P.G., G.T.A. and L.T.D.; Writing-original draft preparation: T.E.M.; Writing-review and editing: T.E.M., P.G., L.T.D. and G.T.A. All authors have read and agreed to the published version of the manuscript.

**Funding:** This research received no external funding.

**Data Availability Statement:** All the sentinel products and the GEE code used to process the S2 and S1 in this study are available in the GEE JavaScript environment. The Auxilary data supporting this study's findings are available from the corresponding author upon reasonable request.

**Acknowledgments:** The authors gratefully thank the anonymous reviewers and the editors whose valuable comments and suggestions have helped improve the quality of this article.

**Conflicts of Interest:** The authors declare no conflict of interest.

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
