# Peer review of "Multi-Temporal Passive and Active Remote Sensing for Agricultural Mapping and Acreage Estimation in Context of Small Farm Holds in Ethiopia"

_land, doi:10.3390/land13030335_

Round 1

Reviewer 1 Report

Comments and Suggestions for Authors

Title:  Multi-temporal Passive and Active Remote Sensing for Agricultural Crop Distribution Mapping and Acreage Estimation in the Context of Small Farms Holds in Ethiopia

Abstract (comments):

The technical description regarding classification approaches seem extensive (However, the authors should mention that this is a supervised classification approach using multiple models). The ration on training/validation might not be necessary in the abstract, since will get it in the methods.

Authors do not discuss issues regarding the “Small Farms Holds”, or why this is important (the firs line is general and somewhat vague). Any further idea regarding this as a conclusion remark?

L24: Specify “optical and microwave data”

Spatial resolutions?

Introduction:

Good work.

I would just suggest the authors to include a line on how their approach might contribute or relate to land management (regional development), or how this would fill a knowledge gap.

Materials and Methods (comments):

L158: Error! Reference source not found… Error! Reference source not found” should address this.

L168-183: Which year did the authors chose for classification? It is not clear which dataset was used, or why these specific sensors were selected. The description on sentinel instruments might not be necessary (I would rather get a reference to that), but more of a description on the dataset actually used is.

L185-196: How many points or polygons did the authors used to train/validate the classifications derived? Please elaborate on the approach.

The classification (class) scheme should be described in this section.

This message “Error! Reference source not found… Error! Reference source not found” keeps appearing in the document.

Results (comments):

This message “Error! Reference source not found… Error! Reference source not found” keeps appearing in the document.

Figures 5 and 6 might not be necessary, since the metrics can be incorporated in a single comparative table. The same can be done with tables 2 and 3 (a comparison might be more useful).

Figures 7 and 8 are redundant. The same happen with 9 and 10.

Figure 11 should be much better explained and constructed, if it needs to be included… there are numerous acronyms, numbers and signs, and the caption does not describe any of the elements. There are no units on it either…

This section is longer than needed. The authors need to edit tables and describe accordingly.

Conclusion (comments):

The conclusion could be well supported by the authors results. However, the results section needs edition.

Author Response

In attachment a letter with a detailed replied to the reviewer's comments.

Kind regards

The authors

Reviewer 2 Report

Comments and Suggestions for Authors

The authors claim that the scope of their research is to address the urgent need to quantifying crop acreage in a timely, accurate, granular and sustainable manner, to the benefit of food security, advancement of the country torwards SDG goals and for the production of official statistics.

The authors explain the issues typically asosciated in the use of optical data for LCLU mapping, in particular the different spectral response recorded for the same features types due to atmopsheric artifacts, and furthermore the issue of cloud cover, leading to low classification accuracy (commission and omission errors). 

As a solution to the above the authors propose to use both Sentinel2 and Sentinel 1, claiming that the combined use will increase class separability and reduce errors.

Finally the uthors apply 4 different supervised methods to peroduce a LCLU map featuring 5 land cover classes and compare both the visual results of the maps, the results from the confusion matrix, and also the comparison of the acreage of the land cover classes in % against official statistics and against global land cover maps.

The authors conclude that the addition of Sentinel 1 to Sentinel 2 increases the accuracy of results across all tests, and in particular RF and the SVM are the two best performing algorythms.

So far the research is robust in its methodology and provides fairly clear explaination of methods and results.

My comments and questions to the authors are the following:

1) could you better explain how you built your EO data sets from which you have extracted the features to train the algoryth,. from the paper it is not clear if you used a time series approach, or a single acquisition approach. If you used a time series approach, like I think, then could you eplain if you mosaiked your data into temporal stacks? which algorythm did you use to impute pixels values?

2) the survey design used to collect the grund truth data would benefit from more explainations. It is claimed that data was gathered both in the field and both from visual interpretation of satellite images. However it is not mentioned the quantity of data collected from each of the two activities, and this is a relevant information. Secondly there is no information about the survey design used to establish the number of samples to be collected, and the allocation of samples per class, and furthermore the sampling unit utilized. These information are very important to understand the robustness of the in-situ data in its capacity to describe the full land cove population. There is no information either on the dates of aquisition of the ground truth data (and that of the satellite images which links to the poin 1 listed above)

3) the overall results show very good accuracy based on the confusion matrix. However these results should be contextualized with the excercise at hand. Despite the title of the paper refers to crop mapping, this excercise is more a land cover mapping one, with only 5 generic land cover classes. In this context I am afraid that the overall results may not be so exciting, especially givent the huge amount of training data (7482) and a relative small area of interest.

4) the extraction of acreage from the maps seems to be based on pixel count. Did you adopt any measure to take into account the commission/omission errors, what is the confidence interval? what is the bias in area estimation?

5) what is the time frame of your study? there is no information about the reference year in the entire paper

Conclusion:

In my view the authors work is more relevant fro LCLU mapping, but less for crop type mapping. I would definatelly encourage that they continue to work on this research agenda, as a second research paper, and try to map the actual crop types, 

Author Response

(The authors gave the same response as above.)
